# Redox Implications of Extreme Task Performance: The Case in Driver Athletes

**DOI:** 10.3390/cells11050899

**Published:** 2022-03-05

**Authors:** Michael B. Reid

**Affiliations:** College of Health and Human Performance, University of Florida, Gainesville, FL 32611, USA; michael.reid@ufl.edu

**Keywords:** oxidative stress, homeostasis, mitochondria, heat stress, reactive oxygen species, nitric oxide, exercise, dehydration, skeletal muscle, g loading, motorsport

## Abstract

Redox homeostasis and redox-mediated signaling mechanisms are fundamental elements of human biology. Physiological levels of reactive oxygen species (ROS) and reactive nitrogen species (RNS) modulate a range of functional processes at the cellular, tissue, and systemic levels in healthy humans. Conversely, excess ROS or RNS activity can disrupt function, impairing the performance of daily activities. This article analyzes the impact of redox mechanisms on extreme task performance. Such activities (a) require complex motor skills, (b) are physically demanding, (c) are performed in an extreme environment, (d) require high-level executive function, and (e) pose an imminent risk of injury or death. The current analysis utilizes race car driving as a representative example. The physiological challenges of this extreme task include physical exertion, g loading, vibration, heat exposure, dehydration, noise, mental demands, and emotional factors. Each of these challenges stimulates ROS signaling, RNS signaling, or both, alters redox homeostasis, and exerts pro-oxidant effects at either the tissue or systemic levels. These redox mechanisms appear to promote physiological stress during race car driving and impair the performance of driver athletes.

## 1. Introduction

Over the past half a century, redox stress and redox-mediated signaling mechanisms have emerged as fundamental elements of human biology. Much has been learned about the generation of reactive oxygen species (ROS) and reactive nitrogen species (RNS), their roles in signal transduction and cellular adaptation, and the antioxidant processes that buffer these redox cascades. For obvious reasons, much of this research has focused on redox mechanisms that impair human health in conditions such as aging, acute inflammation, and chronic disease. Less is known about redox mechanisms that modulate physiological processes and can limit the performance of healthy individuals in their daily activities.

Redox limitations to human performance have been demonstrated in laboratory studies of isolated muscle groups, whole-body exercise, and cognitive tasks. By extension, redox mechanisms are thought to also limit the spontaneous performance of more complex activities in a variety of settings. Examples include athletic competition and activities in the workplace. This article considers a special condition that bridges these two fields, i.e., the risks posed by redox stress when performing extreme tasks.

For the purposes of this article, an extreme task is defined by five characteristics: the task requires one or more complex motor skills; the task is physically demanding; it is performed in an extreme environment; successful execution requires high-level executive function; and finally, the task poses an imminent risk of serious injury or death. Extreme tasks are routinely performed by workers in various professions. Examples include war fighters, commercial divers, fighter pilots, astronauts, fire fighters, race car drivers, and alpine rescue teams.

Redox limitations to extreme activities are largely undefined. In part, this reflects the challenges of such research. Collection of primary data is difficult during extreme activities, and laboratory models are imperfect. Accordingly, there are few systematic studies of extreme task performance, and mechanistic research is largely nonexistent. For these reasons, we have little direct knowledge of redox physiology during extreme tasks or the limitations posed by redox mechanisms. Still, a conceptual framework can be used to predict redox limitations by integrating the known traits of a given task with the broader understanding of redox biology.

The present article develops such a framework. For this exercise, race car driving was selected as the representative activity. There are several reasons for this: First, race car driving typifies an extreme activity, meeting each of the criteria stated above. Second, the physiological data needed to define a conceptual framework are documented in the archival literature [1]. Third, racing is reported to alter the redox status of driver athletes [2]. Lastly, race car driving is a widely publicized activity that will be familiar to most readers.

The physiological challenges faced by driver athletes include the physical work of car control (e.g., steering, braking), g-loads, vibration, noise, heat exposure, and mental factors, e.g., cognitive demand, discomfort/pain, emotion, etc. Drivers perceive and adjust to these demands as they race—a classic example of feedback control. An applied model of feedback control is illustrated in Figure 1. This model depicts the basic relationships among the physiological processes that are used in race car driving and are susceptible to driving-related challenges.

Over time, the challenges of racing promote negative outcomes in the driver. These include dehydration, elevated metabolism and cardiovascular demand, fatigue, mental errors, performance loss, and greater risk of accident/injury. The mechanisms that lead to these outcomes appear to have one trait in common: in each case, the challenge stimulates production of ROS or RNS, which mediate negative effects on the driver. Subsequent sections will address the physiological challenges of race car driving, the effects they have on driver athletes, and the redox mechanisms that appear to be involved. Table 1 provides an overview of the pathways thought to regulate these responses.

## 2. Physical Work

The physical demands of race car driving are substantial. First, car control during competition requires repeated, high-intensity work by specific muscle groups over long periods of time. For example, engineering assessments of driver input illustrate the forces required to drive a high-level race car on a road course during a race that lasted 98 min [3]. Forces developed by the driver during heavy braking averaged 600 N per turn—a one-legged task repeated 225 times. Steering effort averaged 157 N per turn—a task performed 1105 times using arm and shoulder muscles. It is no surprise that professional drivers have identified strength of the limbs and upper body as the major physical demands of their profession [4].

Second, drivers must oppose gravitational (g) forces generated by the race car during changes in direction or velocity. In the horizontal plane, g force along the lateral axis is proportional to the radius of curvature and velocity of a turn. Braking or acceleration develops g force along the longitudinal axis; such forces are proportional to the rate of change in velocity. Modern race cars that are optimized for aerodynamic downforce can exceed 5 g during cornering and braking [5]—physical loads that are applied to the driver.

The physical effort required to resist g loads is among the greatest demands on a race car driver. For example, a driver’s head and race helmet combined weigh approximately 6.4 kg. During heavy braking, the axial loading applied to the head and helmet is effectively 259 N—a force counteracted by the muscles of the neck and upper torso that oppose neck flexion. Moreover, during cornering and braking, drivers use an anti-g straining maneuver developed by fighter pilots [6], in which muscles of the trunk are tensed and the breath may be held [3,7]. This maneuver stabilizes body posture, protects the abdominal viscera, and preserves cerebral blood flow at the expense of muscular effort.

Strenuous exertion places substantial metabolic demand on the driver. This is estimated to range from 5.3 to 13.0 metabolic equivalents, or METs [5,8,9], which is comparable to MET values for traditional sports, e.g., basketball 8.0, soccer 10.0, and boxing 12.8 [10]. Direct measurements of oxygen consumption (VO_2_) by professional race drivers showed that sustained VO_2_ averaged 2.76 L·min^−1^ while driving a typical road course; this was 79% of the maximal VO_2_ measured in the same drivers under laboratory conditions [11]. Separate measurements of blood chemistry under race conditions have shown robust increases in free fatty acids [12,13], triglycerides [13], and glucose [12,14]—circulating substrates that fuel aerobic metabolism. Longer periods of continuous driving cause an increase in blood lactate levels [12,14,15,16]—further evidence of metabolic demand. Elevated metabolism also increases heat production via aerobic metabolism and blood flow to working muscles; these two factors are discussed in greater detail below.

Fatigue is among the greatest challenges for driver athletes [4]. Over the course of a race, the work required to control the car and resist g forces causes the muscles of the arms, shoulder girdle, neck, trunk, and legs to fatigue [15]. This affects steering, braking, and postural control, impairing the performance of the driver. The degree of fatigue varies between muscles and is time- and task-specific. Fatigue is negligible in events of short duration—e.g., drag or sprint races—but can be a major concern in longer races. The race format also matters. Road races require frequent changes in direction and speed, requiring more work and promoting greater fatigue than speedway races of similar duration. Secondary factors that contribute to driver fatigue include vibration of the car and its controls [17], heat exposure [18], and prolonged mental challenges [19], as discussed in subsequent sections.

In aggregate, drivers perform repetitive, forceful contractions by major muscle groups of the limbs, neck, shoulder girdle, and trunk. This intense, whole-body exercise can continue for up to four hours. During this time, the working muscles exhibit elevated metabolism, greater heat generation, higher rates of perfusion, and increased oxidant production; the latter is of particular interest for this review article.

Oxidants are generated at increased rates by working muscles. Bjugstad et al. [2] have shown that racing induces oxidative changes in driver athletes, shifting the redox potential of mixed venous blood to a more oxidized state. This response was related to pre-race antioxidant capacity. Redox status was less perturbed in drivers who had higher antioxidant capacities and those who took vitamin supplements. In part, the pro-oxidant shift caused by racing reflected an increase in oxidant production by working muscles. Skeletal muscle continually generates ROS, which are detectable within myofibers [20] and in the extracellular space [21]. The parent molecule of the ROS cascade is the superoxide anion radical, which gives rise to hydrogen peroxide, hydroxyl radicals, and other low-molecular-weight derivatives. Sources of ROS in differentiated myofibers include the mitochondrial electron transport chain, nicotinamide adenine dinucleotide (phosphate) oxidase—or NAD(P)H oxidase (NOX)—xanthine oxidase, cyclooxygenases, and lipoxygenases. Among these, ROS from specific oxidases appear to modulate cell signaling via redox-sensitive regulatory proteins [22], whereas ROS that modulate the intracellular redox environment derive from both mitochondria [23] and NOX [24].

In parallel, skeletal muscle continually generates nitric oxide, or NO [25]. This may be produced via either of two constitutively expressed NO synthases: a tissue-specific isoform of neuronal-type nitric oxide synthase—or nNOS [25]—and the endothelial-type NO synthase, or eNOS [26]. NOS-independent synthesis of NO from nitrate may also be possible under certain conditions [27]. NO and its redox derivatives compose the RNS cascade. NO can act on target proteins via cGMP-dependent signaling or via direct redox modification, e.g., S-nitrosylation. By these mechanisms, NO modulates various aspects of skeletal muscle function, including contractile function [25,28], mitochondrial respiration [29], and glucose uptake [30].

Both ROS and RNS activities increase in muscles during repetitive contractions [20,25], such as those that occur in the limb and trunk muscles of driver athletes during competition. Increases in ROS and RNS activities shift the intracellular redox status, promoting oxidation of regulatory proteins and altering myofiber function in a time- and concentration-dependent manner. Increases in oxidant activity can disrupt excitation–contraction coupling [31], calcium sensitivity of myofilament proteins [32,33], mitochondrial energetics [34], and carbohydrate metabolism [30]. These intracellular events combine to depress the specific force of muscle contraction [35] and contribute to muscle fatigue [36]—a near-ubiquitous finding in the muscles of driver athletes following a race [15].

## 3. Vibration

Race cars vibrate. The sources of vibration include rotation of drivetrain components, wheel-and-brake assemblies, interactions between the tires and the road surface, flexing of aerodynamic surfaces, and harmonics of the chassis, engine, and other components [37,38,39]. The race car structure is stiff, and there is little to dampen the transmission of vibrations to the driver [40,41,42]. Vibrations transmitted to the driver act as both whole-body inputs and localized mechanical stimuli, eliciting distinct physiological responses in each case.

Whole-body vibration has multiple effects on drivers. At high speeds, interactions of the car with variations in the track surface can impose transient vertical loads of up to 3 g on the driver [42]. Vertical loading can cause substantial discomfort and injury in drivers. Indeed, a retrospective study of musculoskeletal injuries in over 130 drivers identified pain in the lumbar back region and upper legs as one of the most common complaints [43]. Whole-body vibration can slightly impair driver vision, although this appears to be a transient phenomenon with no serious consequences. Laboratory studies of healthy adults show that whole-body vibration stimulates reflex increases in heart rate, cardiac output, oxygen consumption, and minute ventilation [44]. These changes mimic the cardiopulmonary responses seen in drivers during competition [1]. Similarly, whole-body vibration causes mental fatigue and diminishes mental alertness in noncompetitive driving [17]. Such issues affect race drivers during long stints, especially in cross-country rallies and endurance racing, and may be caused in part by vibration.

Localized transmission of vibrations to the driver via the steering wheel is a well-defined problem [45]. It predisposes muscles of the hands, arms, and upper body to cramping and fatigue, which can impair car control and distract the driver. Steering wheel vibrations are also linked to symptoms and nerve disorders of the hands and arms [43]. This is consistent with mechanistic research showing that vibration of an appendage affects the associated peripheral nerve. Acute vibration sensitizes nerve fiber populations that carry pain and proprioceptive information—notably Aβ, Aδ, and C fibers [46]. Over longer time periods, repetitive bouts of vibration cause loss of small nerve fibers, increase mitochondrial numbers in fibers, and increase the size of sensory receptors [47]. Adaptations in the afferent pathway are associated with pro-inflammatory changes that include cytokine upregulation and oxidative stress [46].

Vibration also elicits redox responses. Circulating markers of oxidative stress, as documented in driver athletes [2], are among the earliest biochemical indicators of vibration-induced disease in humans [48]. These include changes in the serum activities of superoxide dismutase (SOD) and catalase—enzymes that selectively degrade superoxide anions and hydrogen peroxide, respectively. Mechanistic studies show that selective vibration of a single appendage has widespread effects on redox status. Increased ROS levels have been documented in the appendage skin, arterial walls, and sensory receptors—responses that are sensitive to vibration frequency [47,49]. Appendage vibration also induces oxidative stress in remote tissues, including the dorsal root ganglia that serve the appendage [46] and the heart [50]. Redox adaptation to vibration is suggested by alterations in mRNA levels for nNOS and antioxidant enzymes in the heart, eNOS and oxidative stress genes in the prostate, and pro-inflammatory cytokines in multiple tissues [50].

Experiments have shown that physiological responses stimulated by vibration are sensitized by direct ROS exposure and blunted by selective ROS depletion [51]. This provides proof-of-concept that ROS can play a causal role in the vibration-related problems experienced by race car drivers. Specifically, oxidative stress promotes neuroinflammation [52,53] which mediates lower back pain [54,55]. Such pain is a chronic malady that plagues race car drivers and is caused by whole-body vibration. Similarly, steering wheel vibration causes localized pain, numbness, and cramping in driver forearms. These symptoms are predicted to result from the localized effects of oxidants on peripheral nerves and dorsal root ganglia [56], as well as on muscle spindles [57].

## 4. Noise

Environmental noise has deleterious effects on human health. Brief periods of noise exposure can interfere with work performance [58] and promote hearing loss that may be irreversible [59]. Repeated noise exposure or continuous long-term exposure can have non-auditory health consequences; these include sleep disturbance [60], impaired cognition [61], circadian dysfunction [62], vascular dysregulation [63], and cardiovascular disease [64]. The severity of such risks has led government agencies to establish standards for occupational noise exposure in numerous countries. For example, United States standards are overseen by both the Occupational Safety and Health Administration (OSHA) and the National Institute of Occupational Safety and Health (NIOSH) [65].

Noise is a prominent feature of auto racing. Driver athletes continuously work near unmuffled race cars and noisy support machinery such as generators, jacks, and compressors. Ambient noise at race tracks can reach peak levels of 130–140 dB [59,66,67], exceeding the exposure limits recommended by OSHA and NIOSH [65,66]. Accordingly, noise has deleterious effects on drivers, interfering with communication [68,69] and potentially impairing performance [58]. Hearing loss is a ubiquitous risk of the profession and a common finding among experienced drivers [66]. Current research predicts that drivers are also at risk of non-auditory health effects of noise on sleep quality, cognitive ability, and cardiovascular regulation. Noise may also promote inner ear balance disorder—a consequence of noise-induced inflammation of the vestibular system. The prevalence and severity of non-auditory noise effects on driver athletes has not been reported.

Noise-induced hearing loss is caused by damage to the cochlea—a delicate structure of the inner ear. Cochlear hair cells transduce mechanical energy to electrical energy, converting sound into afferent neural input that can be interpreted by the brain [70]. Animal experiments show that loud noises damage key components of this sensory pathway, including stereocilia and hair cells [71], hair cell synaptic ribbons [72], spiral ganglia neurons [73], and the stria vascularis [74]. Functional results of this damage include elevated hearing threshold [75], disorders of speech perception and auditory processing [76], and tinnitus [77], all of which are common among driver athletes [66,68,69].

Oxidative stress mediates the damage to auditory structures caused by noise [78]. Noise at damaging levels stimulates ROS activity in the cochlear tissue [79]—a response that precedes cellular injury and persists for a week or more [80]. Evidence of oxidative stress includes increased lipid peroxidation products and protein nitrosylation [81], as well as altered glutathione regulation [82,83]. Importantly, noise-induced cochlear damage is blunted by pretreatment with nonspecific antioxidants, including vitamin E and alpha lipoic acid [84], ebselen [85], glutathione [86], and N-acetylcysteine [87]. Specific evidence for ROS involvement derives from experiments showing that noise-induced damage can be inhibited by ROS-selective probes, such as either exogenous SOD [88] or hydroxylated alpha-phenyl-tert-butylnitrone—a nitrone-based spin trap for hydroxyl and superoxide anion radicals [89]. Mitochondria are the primary source of noise-induced ROS in cochlear tissue [90]. The increase in ROS production is attributed to a rise in aerobic metabolism [90] and loss of mitochondrial membrane potential [75]. These changes drive electron leakage from the electron transport chain, thereby stimulating superoxide anion formation. NOX-derived ROS also contribute to noise-induced cochlear damage. This follows from observations that NOX subunit expression is altered by damaging noise [91], signaling via the NOX3 isoform mediates noise-induced inflammation [92], and NOX inhibition reduces the damage caused by noise exposure [93]. In aggregate, the science indicates that hearing loss, difficulties in speech perception, and tinnitus experienced by driver athletes are the result of cochlear damage caused by repeated bouts of oxidative stress mediated by noise-induced ROS production.

Beyond the acute effects of individual race events, driver athletes live in a professional environment that predisposes them to persistent noise exposure. This is a major environmental stressor that promotes mental stress, disrupts sleep, and is associated with cardiometabolic risk factors [60,64]. Unlike hearing loss, the non-auditory responses to persistent noise are not attributed to a primary unifying mechanism. Rather, the existing research links persistent noise exposure to pathogenic risk factors that include chronic elevation of stress hormone levels and markers of inflammation, e.g., circulating cytokines, pro-inflammatory signaling, nitric oxide synthesis, and ROS activity [64,66,79].

## 5. Heat

Race cars generate large amounts of thermal energy during competition. Heat from the engine, exhaust system, brakes, and tires combine to warm the cockpit of the car. Cockpit air temperatures of 50–60 °C are common, with solid surfaces reaching 100 °C [5,18,94,95]. In this hot environment, the driver athlete continuously works at 65–85% of maximal VO_2_ for up to four hours while wearing a helmet, gloves, and multilayered fire-resistant safety clothing. In combination, these factors create the conditions for uncompensable heat stress [96]. Skin temperature rises, core temperature rises faster, and the core-to-skin temperature gradient for heat dissipation falls progressively [97]. Sweating is profuse but ineffective because evaporative cooling is prevented by the safety clothing. As a result, drivers routinely lose 3–4% of body weight due to dehydration, with corresponding increases in hematocrit and urine osmolarity [14,18,98]. Heat stress and dehydration combine to impose thermal and physiological strain on the driver [97,99,100,101] that can increase the perception of exertion and discomfort [101], diminish performance [18,99], impair cognition [102], cause heat stroke [94,95], and induce loss of consciousness [103]. Heat exposure is amplified in closed-cabin cars, hot weather, and prolonged stints of continuous driving [5,18]. Drivers can mitigate the effects of heat through a program of acclimatization [99], and by in-car use of cooling technologies [95,100].

Oxidative stress is associated with environmental heat exposure in various occupations [104,105,106,107,108,109], including athletics [104,110,111,112,113]. This is consistent with data from driver athletes. Drivers exhibited increases in body temperature and oxidative stress after racing under warm conditions (28.3 °C), as compared with baseline measurements under cooler conditions (21.3 °C) [2]. McAnulty et al. [104] first demonstrated that hyperthermia increases oxidative stress during exercise—a robust finding that has been confirmed by others [113,114]. Thus, exercise and environmental heat have additive effects on redox status, amplifying pro-oxidant changes at the systemic level [115].

To define the underlying physiology, researchers have tested the interactions between physical activity, heat, and dehydration in healthy volunteers, measuring plasma biomarkers to assess changes in redox status. Hillman et al. [116] showed that pro-oxidant changes caused by hyperthermic exercise were exaggerated if subjects became dehydrated. This confirms the risk to driver athletes, who routinely become dehydrated during competition. Laitano et al. [117] established that hydration state modulates glutathione regulation by muscles. Glutathione is normally taken up by muscles, whereas dehydration stimulates exercising muscles to release glutathione into the circulation. Consistent with this finding, Georgescu et al. [118] found that dehydration per se alters plasma antioxidant capacity in the absence of heat exposure or exercise. Redox perturbations caused by heat exposure or exercise appear to be mitigated by exercise training [113] and may be acutely reversed by rehydration [119], although the latter finding remains controversial [117,118].

Heat exposure induces oxidative stress at the cellular level by increasing ROS production and decreasing antioxidant protection. Mitochondrial production of superoxide anion radicals is stimulated by damage to the electron transport chain [120] and by a reduction in the synthesis of uncoupling proteins [121]. Generation of superoxide anions by NOX is also stimulated due to upregulation of the NOX1 isoform and an increase in the cytoplasmic ratio between oxidized nicotinamide adenine dinucleotide phosphate (NADP+) and reduced NADP+ or NADPH [122]. In parallel, heat accumulation compromises the oxidant buffering capacity of the cell by inactivating mitochondrial SOD [123] and depleting glutathione levels [124].

King et al. [125] have extended this model to include the effects of exercise and dehydration in vivo. These authors acknowledge the traditional effects of heat on cellular redox status (above), but further observe that evaporative water loss and sweating during hyperthermic exercise leads to hemoconcentration and increased blood viscosity. The rise in blood viscosity increases shear stress on vessel walls. This stimulates endothelial release of nitric oxide [126] and ROS [127] that act on erythrocytes to increase rigidity and impair deformability [128,129]. These biophysical changes alter the rheological properties of blood, promoting hemolysis. The subsequent release of free hemoglobin promotes further ROS generation via Fenton-like reactions, a chemistry catalyzed by the ferrous iron in heme centers.

Redox responses to whole-body heat exposure are well documented. Studies indicate that heat accumulation preferentially affects mitochondria-rich tissues, including the heart, liver, and aerobic skeletal muscles. Thus, heat exposure preferentially targets tissues stressed by the activity of race car driving, imposing an additional physiological load on heavily working skeletal and cardiac muscles. Heat exposure acts on these tissues to disrupt redox homeostasis and mitochondrial function. Malondialdehyde, a marker of lipid peroxidation, is elevated in aerobic muscles following heat accumulation [130,131]. This is accompanied by increased tissue content of SOD mRNA and SOD proteins, as well as elevated SOD enzyme activity [130,132]. Heat stress also alters catalase content and enzyme activity in aerobic muscles [130,132]. Mitochondrial protein analyses showed that cytochrome c, cytochrome c oxidase, voltage-dependent anion channel, pyruvate dehydrogenase, and prohibitin 1 were all elevated [130]. In a separate study [131], biochemical analyses of subsarcolemmal mitochondria confirmed that malondialdehyde levels are elevated following heat exposure and that carbonylation of mitochondrial proteins is widespread—further evidence of oxidative stress in these organelles.

## 6. Cardiovascular Demand

Cardiovascular regulation is linked to racing-related changes in oxidant activity [2] and is among the best documented aspects of driver physiology. For over half a century, we have known that driver heart rate is elevated before a race begins [133]—an anticipatory response stimulated by increases in circulating epinephrine and norepinephrine [134]. Heart rate remains elevated throughout the event, with sustained heart rates of 65–90% of maximal being common [95,134,135]. Impedance cardiography under race conditions has shown that stroke volume is also increased [136], further contributing to a rise in cardiac output. In general, heart rate increases with the average speed of the car on a given track [11]. However, instantaneous heart rate varies according to driver position on the track, being higher in corners and lower on straight sections [137]. Similarly, average heart rate is greater during races on road courses than races on speedways, despite the latter having higher average speeds [11]. Thus, speed per se is not the sole determinant of heart rate; the physical work of turning and braking is also important. Cardiovascular function is further influenced by positive g loading in the vertical axis, which promotes blood pooling in the lower extremities. Prolonged exposure can limit venous return to the heart, thereby depressing cardiac output and systemic arterial pressure, which can cause cerebral hypoxia. These are real concerns for drivers racing on speedways, where banked corners impose significant vertical g loads. In extreme cases, this can cause visual disturbances and even loss of consciousness in drivers [138]. A final contributor to cardiovascular stress is heat, which elevates sweat rate, blood flow to the skin, heart rate, and cardiac output for a given level of physical work [139]. This thermoregulatory reflex is pervasive in driver athletes across a broad range of racing series [97,101,140]. Prolonged heat exposure promotes dehydration—a common risk for driver athletes [141,142]. Dehydration promotes cardiovascular stress by decreasing the volume of circulating plasma and blunting heart rate variability [143].

The cardiovascular responses of driver athletes are consistent with the acute changes in cardiac function mediated by redox signaling. The contractile function of the healthy heart is modulated by endogenous ROS generated by mitochondria and NOX. Cardiac myocytes produce ROS in proportion to contractile activity and oxidative metabolic rate [144,145]. These are generated by electron leakage from complexes I and III of the mitochondrial electron transport chain [146]. This basal rate of ROS production does not appear to influence contractile function [147,148]; however, endogenous ROS do modulate contraction as second messengers in two parallel pathways.

First, the “fight-or-flight” response seen in driver athletes [149] is mediated in part by sympathetic stimulation of β-adrenergic receptors in the myocardium. This increases the force and frequency of heart contractions, elevating cardiac output. At the cellular level, β-adrenergic stimulation increases endogenous ROS production as well as contractility [150]. Mitochondria and NOX are both implicated as sources of β-adrenergic-stimulated ROS in this response [147]. Potential downstream target(s) of ROS include redox-sensitive kinases, phosphatases, and regulatory proteins of the myofilament lattice, sarcoplasmic reticulum, or sarcolemma [151]. Studies using antioxidant probes indicate that endogenous ROS can either oppose [147,148] or amplify [150] the contractile effect of β-adrenergic stimulation. The net effect may depend on the source of endogenous ROS, the subcellular localization of the source, and the response of the adenylyl cyclase-cAMP-PKA pathway [151].

Second, endothelin-1 (ET-1) is a potent vasoconstrictor that regulates blood pressure under conditions experienced by drivers, including prolonged exercise [152] and altered g loading [153]. ET-1 is constitutively expressed by the cardiac endothelium, vascular smooth muscle, fibroblasts, and myocytes. Among its effects on the heart, ET-1 is an important regulator of myocardial contractility. ET-1 binds to surface receptors, activating a redox-mediated signaling cascade that increases the force of contraction. Cytosolic signaling events in the ET-1 pathway include NOX production of superoxide anion radicals [154]. Superoxide anions activate extracellular signal-regulated kinase 1/2 (ERK1/2), which acts via p90 ribosomal S6 kinase (p90RSK) and sodium/hydrogen exchanger-1 (NHE1) to increase myofilament calcium sensitivity [147], thereby increasing the force of contraction.

Like skeletal muscle, cardiac myocytes constitutively express both nNOS and eNOS, which have distinct subcellular localizations [155]. The nNOS isoform localizes to the sarcoplasmic reticulum (SR) and plays an active role in β-adrenergic receptor signaling [156]; nNOS acts via guanosine 3′,5′-cyclic monophosphate (cGMP) to stimulate phospholamban phosphorylation, which disinhibits calcium uptake by the sarcoplasmic reticulum (SR) calcium ATPase, or SERCA [157]. This increases SR calcium stores, calcium release, and cardiac contractility in response to β-adrenergic stimulation, contributing to the rise in stroke volume and cardiac output in driver athletes under race conditions [136]. In contrast, eNOS localizes to the caveolae of the transverse tubule system, and acts on the L-type calcium channel to limit the inward calcium current. eNOS signaling exerts negative effects on β-adrenergic receptor signaling, blunting calcium release and depressing cardiac contractility [158], as well as limiting arrhythmogenic activity [159].

## 7. Mental Factors

The physiological basis for elite performance by driver athletes is grounded in motor control and brain neurobiology. Race drivers differ from non-racing drivers in their execution of task-specific motor skills; oculomotor behavior is a notable example. Unlike non-racing drivers, race drivers use head direction to determine gaze angle, which is tightly linked to steering angle and vehicle rotation—a behavior that appears to be controlled at the unconscious level [160]. Moreover, their driving behavior differs from that of non-racing drivers. Race drivers drive a different path when negotiating a racetrack and are more decisive in throttle and brake applications [161]. Furthermore, their gaze strategy is more variable, and there is greater head rotation while turning the vehicle, which led Land and Lee [162] to conclude that race drivers have different perceptual–cognitive skills than non-racing drivers.

Behavioral differences are complemented by differences in brain function and structure. Bernardi et al. [163] used functional magnetic resonance imaging (fMRI) to compare the brain function of race and non-racing drivers while performing standardized mental tasks. For a given task, race drivers were found to recruit smaller volumes of the task-related brain regions and exhibit stronger functional connections between regions. From these results, the authors concluded that that race drivers benefit from greater “neural efficiency”. A second study by the same research team [164] used functional magnetic resonance imaging (fMRI) to measure brain function and structure while subjects watched videos of cars racing on tracks. Race drivers were found to recruit a larger number of brain regions than non-racing drivers, and more consistently recruited the brain regions related to motor control and spatial navigation. Race drivers also had a greater density of gray matter in driving-related brain regions, including the retrosplenial cortex. Among race drivers, gray matter density of the retrosplenial cortex was directly correlated with success in the sport, as reflected by top-three finishes in prior races. The results of these fMRI studies are consistent with the assertion that elite race drivers have unique brains, functionally and structurally different from non-racing drivers [165].

Elite brain notwithstanding, the mental demands of racing are an omnipresent challenge that can affect drivers’ emotions, cognition, and performance [166]. Changes in mental state can be inferred, in part, by activity of the sympathoadrenal axis. Moment-to-moment changes are reflected in driver heart rate. Anticipation causes heart rate to increase in the moments before a race begins [137], imminent danger stimulates heart rate during a crash [167], and competition per se elevates heart rate compared to practice sessions or reconnaissance runs at the same speed [135,140]. Long-term stress responses are evident in measurements of circulating hormones. Total catecholamine levels are elevated during competition [12]. This response surges at the start of a race, slowly declines over time, and is dominated by norepinephrine [13]. Other stress hormones elevated during racing include cortisol [19], aldosterone [16], testosterone [138], and human growth hormone [12].

Environmental and occupational factors can further amplify the problem. Environmental noise interferes with communication [58], promotes annoyance [61], disrupts sleep patterns [168], and activates the sympathoadrenal axis [63]. Sleep deprivation promotes mental stress [169] and is endemic among driver athletes on race weekends. Sleep duration is constrained throughout the weekend by driver commitments to the team, sponsors, race organizers, media, and fans. During available sleep periods, the quality of sleep may be compromised by task-related distractions and pre-performance anxiety. Sleep deprivation is particularly acute during endurance races, which last up to 24 h, or cross-country rallies that continue for days. Finally, international travel in some elite racing series can disrupt circadian rhythm—yet another factor promoting mental stress [170].

What are the redox implications of mental stress? In 1996, an early review by Moller et al. [171] concluded that evidence linking psychological stress to oxidative stress was “rudimentary”, citing only five references. How the field has changed! A PubMed search in January 2022 using the terms “mental stress” and “oxidative stress” returned over 3900 archival reports—the result of exponential growth in published research over the past 35 years. This expansive literature confirms that mental stress impacts redox homeostasis across a broad array of conditions. In humans, circulating biomarkers have been used to screen for oxidative stress under conditions of mental stress, repeatedly demonstrating elevation of circulating oxidation products [172,173,174,175] and decrements in antioxidant status [174,175,176]. Animal models of stress confirm changes in systemic biomarkers [177,178,179]; they further demonstrate oxidative modifications to brain regions that are involved in the stress response, including the hypothalamus [179,180] and brainstem [181]. Driver athletes routinely cope with mental challenges that impact redox status, including strenuous physical activity [174,182,183,184], environmental noise [64], sleep deprivation [169], fatigue [19], and circadian dysregulation [62].

The mechanisms responsible for oxidative stress in acute mental stress are not well defined. Regional effects within the brain likely reflect transient increases in the local activity of neuronal mitochondria and constitutively expressed nNOS or NOX isoforms. At the systemic level, a common denominator across many forms of acute mental stress is the neuroendocrine stress response; this is mediated by central activation of two complementary pathways, both of which are activated during race car driving [12,13,19]. The sympathoadrenal axis is initiated by brain input to sympathetic preganglionic neurons located in the thoracic spinal cord. These neurons respond to excitatory input by stimulating the release of catecholamines—i.e., norepinephrine and epinephrine—via two routes. Preganglionic stimulation of postganglionic neurons causes norepinephrine release from nerve endings in target tissues. Stimulation of chromaffin cells in the adrenal medulla causes the release of norepinephrine and epinephrine into systemic circulation. In contrast, the hypothalamic–pituitary–adrenal (HPA) axis is activated by stress-related glutamatergic input to neuroendocrine neurons in the paraventricular nucleus of the hypothalamus. Once activated, the neuroendocrine neurons release corticotropin-releasing hormone (CRH) and vasopressin—hormones that act on the anterior pituitary to stimulate the release of adrenocorticotropic hormone (ACTH). ACTH is transported by the systemic circulation to the adrenal cortex, where it stimulates the release of glucocorticoids, either cortisol in humans or corticosterone in rodents.

The sympathoadrenal and HPA axes are generally co-activated by conditions that cause mental stress [185], and oxidative stress can be induced by the end products of either axis: catecholamines [186] or glucocorticoids [187]. The mechanisms by which these hormones promote oxidative stress are complex and incompletely understood. However, in response to various mental stressors, there is a strong association between oxidative stress and higher rates of mitochondrial ROS production [188,189,190]. This is consistent with actions of the sympathoadrenal and HPA axes. Catecholamines stimulate oxygen demand in vivo [191], increasing aerobic metabolism via the mitochondrial electron transport chain. Glucocorticoids elevate glucose and free fatty acid concentrations in the circulation, providing additional substrate availability for mitochondrial respiration. Elevated activity of the mitochondrial electron transport chain promotes generation of superoxide anions and superoxide-derived ROS, increasing oxidant activity within minutes-to-hours [192]. Changes in the redox environment of target tissues appear to modulate glucocorticoid receptor signaling. Elevated oxidant levels downregulate expression of the glucocorticoid receptor [193], interfere with the dissociation of heat shock proteins from the receptor in the cytoplasmic compartment [194], and inhibit nuclear internalization of the receptor [194]. This blunts the downstream response of target cells to HPA stimulation at the expense of redox homeostasis, predisposing the target tissue to oxidative stress.

## 8. Conclusions

Extreme task performance, as illustrated by race car driving, requires an individual to overcome a complex array of physiological challenges. Some are intrinsic to the task; others are imposed by the environment in which the task is performed, while others reflect secondary demands on homeostatic regulation and mental function of the individual. A common property of these challenges is their potential to alter redox homeostasis. Challenges associated with race car driving stimulate the production of ROS or RNS, depress antioxidant capacity, or both [2]. These responses promote oxidative or nitrosative stress that may be localized to specific tissues, including the skeletal muscle, cardiac muscle, brain regions, or the cochlea. Alternatively, whole-body stimuli such as heat, vibration, and sympathoadrenergic activation have pro-oxidant effects that are distributed systemically.

Integrating these individual responses, one may conclude that challenges associated with race car driving disrupt redox homeostasis and promote four types of physiological stress: muscular stress, cardiovascular stress, heat stress, and mental stress. This is illustrated by the integrative model shown in Figure 2, which depicts driving-related challenges that promote each type of physiological stress: First, muscular stress is promoted by the work of steering and braking required for car control, postural work to resist g loading, increased metabolism of working muscles, vibration effects on peripheral nerves and joints, and dehydration effects on muscle blood flow. In combination, these factors can impair driver performance by causing muscle fatigue, muscle cramping, or both. Second, cardiovascular stress is promoted by increased metabolic demand for oxygen and substrate; elevated heart rate, stroke volume, and cardiac output; increased skin blood flow for heat exchange; reflex responses to vibration, noise, anxiety, and discomfort; decreased venous return during vertical g loading; and hemoconcentration and hemolysis secondary to dehydration. If cardiac reserve is insufficient to meet these challenges, perfusion limitation compromises the physical performance, heat tolerance, and mental acuity of the driver. Third, heat stress is caused by muscular heat generation and environmental heat exposure, compounded by the insulating effects of head-to-toe safety clothing. These factors promote heat retention by the driver, increasing core temperature and driver discomfort. In response, thermoregulatory reflexes increase sweat rate; this promotes driver dehydration, which compounds the deleterious effects of heat per se. Finally, mental stress can result from the complex cognitive demands of high-speed driving; the emotional responses to competition, e.g., anxiety, anger, fear, etc.; the discomfort or pain caused by loud noise, sustained vibration, and intense heat; and the loss of mental acuity promoted by physical exhaustion, sleep deprivation, and circadian dysregulation. In aggregate, these factors predispose driver athletes to mental errors of motor control, spatiotemporal judgement, and executive function that worsen performance on track and make accidents more likely.

Driver athletes are affected by all four physiological stresses—muscular, cardiovascular, heat, and mental—any of which may impair performance. Consider muscle cramps that impede steering, or heat stroke, or a mental error that causes a crash—each reflects overt task failure caused by one type of stress. The more common scenario is more nuanced. Drivers complete races but do not win, admitting that they could have done better. Such subpar performance likely reflects multiple stresses acting in parallel. Each exerts pro-oxidant effects on redox homeostasis, promoting oxidative stress that impairs different aspects of the driving task. Thus, redox disruption represents a common mechanism by which physiological stresses oppose optimal task performance.

It follows that nutritional strategies or training regimes that promote redox homeostasis might improve performance. Consistent with this hypothesis, vitamin supplements appear to reduce racing-induced oxidative stress [2], and media reports routinely trumpet the antioxidant-rich diets of professional drivers. Research is needed to determine the effect of such interventions on racing performance. The results could yield data-driven recommendations that would benefit driver athletes and the motorsports industry.

## Figures and Tables

**Figure 1 cells-11-00899-f001:**
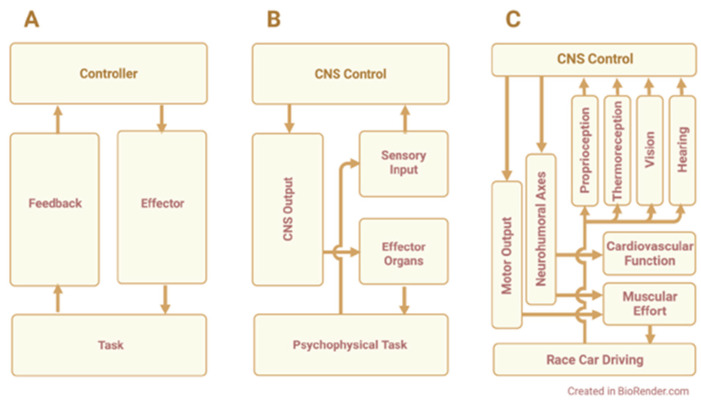
Development of a negative feedback model for race car driving: (**A**) Generic model depicting feedback control; the controller incorporates comparator function, error estimation, correction calculation, and output of control signal; the effector transduces the control signal to perform the task; task performance generates a feedback signal to the controller. (**B**) Biological model under central nervous system (*CNS*) control stimulates effector organs to perform a psychophysical task monitored by sensory feedback. (**C**) Driver athlete model depicts CNS regulation of cardiovascular and muscular function during race car driving, with sensory feedback using proprioception, thermoreception, vision, and hearing. Created in BioRender.com; accessed on 26 January 2022.

**Figure 2 cells-11-00899-f002:**
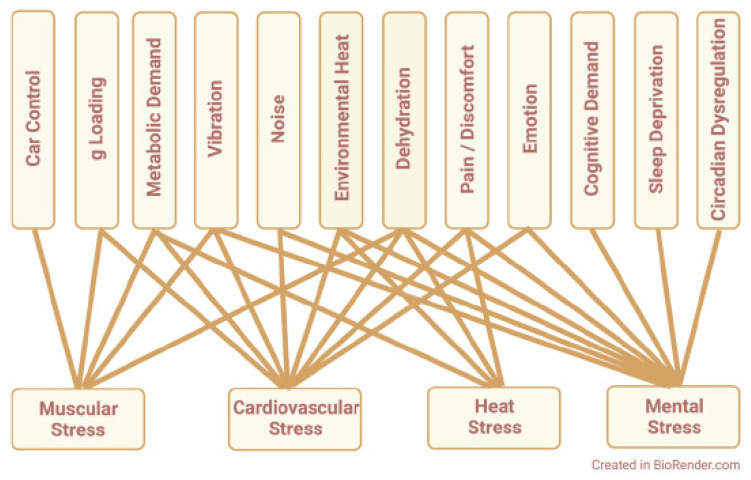
An integrative model of physiological stress caused by race car driving. The model depicts driving-related challenges that act on driver athletes (upper boxes) via redox-mediated pathways (solid lines) to promote distinct categories of physiological stress (lower boxes). Created in BioRender.com; accessed on 27 January 2022.

**Table 1 cells-11-00899-t001:** Proposed pathways for redox signaling in race car driving.

	Physiologic Stimulus	Tissue of Origin	Oxidant Cascade	Intracellular Source(s)	Physiologic Outcome
Pathways for Redox Signaling	physical work, g loading, metabolic demand	skeletal muscle myofibers	ROS RNS	mitochondria, NOX, nNOS	muscle weakness, fatigue
	vibration	skin, blood vessels, sensory receptors	ROS	mitochondria	nerve sensitization, muscle cramps, limb and back pain
	noise	cochlea	ROS	mitochondria, NOX3	tinnitus, hearing loss, sleep disruption
	heat, dehydration	heart, liver, aerobic skeletal muscle	ROS RNS	mitochondria, NOX1, eNOS	antioxidant depletion, cellular damage, hemolysis
	cardiovascular demand	heart, vascular endothelium	ROS RNS	mitochondria, NOX, nNOS, eNOS	altered contractilityand vascular tone
	cognitive load, emotion, discomfort, sleeploss, circadian dysregulation	brain, vasculature, stress hormone-sensitive tissues	ROS	mitochondria	systemic oxidative stress, lucocorticoid desensitization

Details of each pathway with citations are described in Section 2, Section 3, Section 4, Section 5, Section 6 and Section 7; ROS, reactive oxygen species; RNS, reactive nitrogen species; nNOS, neuronal-type nitric oxide synthase; NOX, nicotinamide adenine dinucleotide (phosphate) oxidase or NAD(P)H oxidase; NOX3, NAD(P)H oxidase 3; NOX1, NAD(P)H oxidase 1; eNOS, endothelial-type nitric oxide synthase.

## Data Availability

Not applicable.

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
