# Peer review of "Redox Implications of Extreme Task Performance: The Case in Driver Athletes"

_cells, 2022, doi:10.3390/cells11050899_

Round 1
Reviewer 1 Report
In this manuscript, responses to oxidative stresses caused by race car driving were summarized as a review article. This review describes how various aspects of race car driving influence oxidative homeostasis. These includes muscle contraction, vibration, noise, heat, cardiovascular functions, and mental factors. This review is a comprehensive summary and so would help readers better understand the overview of this field of research. I have several comments as shown below.
1) It was well explained in the manuscript that race car driving has many factors that affect redox homeostasis such as muscle contraction. But each of these are general stimuli that can be observed in many other situations. Since this review is about "the case in driver athletes" as described in the title, it is desired to mention some characteristics specific to driver athletes.
2) There are a lot of unnecessary "-" in the text.
(ex. Line 7, bi-ology.)
3) Line 65. "2. Materials and Methods" should be deleted.
4) Line 231. "glutathione or glutathoine" should be corrected properly.
5) There are unreadable letters.
(ex. Line 173, 346, 348, 373, 375)
Reviewer 2 Report
The narrative review cells-1595013 is interesting and aims to highlight the involvement of oxidative stress in different physiopathological aspects of car drivers. The review shows that various factors including environmental factors can cause a decrease in the physical performance of the pilot. This may also be due to a redox imbalance or an increase in RON/S in different tissues. The review is a thematic study of the work already published: doi: 10.1249 / MSS.0000000000002070. There are several critical issues that do not make the manuscript publishable in this version. Line 75. There is a Materials and Methods title that shouldn't be present. In the introduction section (lines 53 and following) there is no mention of the purpose of the review regarding the involvement of the RON/S. Section 2. Physical effort, there is a good discussion of the RON/S (lines 119 and following) but no reference to the pilots. Section 4. There is no reference to the onset of tinnitus or to the driving disorder induced by the reduction of the sense of balance due to inflammatory phenomena. There are many formatting errors along the text (eg lines 27, 35, 40 etc ... and lack of the font symbol as line 173) which are usually due to transfer of text from different files. Please double check all the text carefully.
Round 2
Reviewer 2 Report
I thank the author for accepting all criticisms and modifying the manuscript accordingly.
Minor revisions:
1) The line numbering of the manuscript differs from that reported in the responses to the reviewer. The changes have been highlighted, so the review is visible even if the control work is made difficult.
2) The author should double-check the numbering of the references as during the revision in some points of the manuscript the numeral sequence of the citations was lost.
3) The figures and tables are repeated at the end of the manuscript. Is it a problem of building the pdf?